DATA RELEASE

# Sampling collections and metadata of planorbidae (Mollusca: Gastropoda) in Brazil: a comprehensive analysis of the Oswaldo Cruz Institute's Mollusk Collection from 1948 to 2023

Silvana Carvalho Thiengo[1], Mariana Gomes Lima[1,*],
Alexandre Bonfim Pinheiro da Silva[1], Raiany Thuler Nogueira[1],
Flávia Cristina dos Santos Rangel[1] and Suzete Rodrigues Gomes[1]

1 Laboratório de Referência Nacional para Esquistossomose – Malacologia, Instituto Oswaldo Cruz, Fundação Oswaldo Cruz (LRNEM/IOC/Fiocruz), Brasil

## ABSTRACT

Planorbidae comprises approximately 40 genera of freshwater gastropods, including roughly 250 species. Among the Planorbidae subfamilies, the significance of Planorbinae is due to its genus *Biomphalaria*, whose species are intermediate hosts of the trematode *Schistosoma mansoni* Sambon, 1907, which causes schistosomiasis in humans and animals. Here, we present the analysis of the dataset of Planorbidae housed in the Collection of Mollusks of the Oswaldo Cruz Institute, with a special focus on *Biomphalaria* species. This dataset includes 7,267 lots originating from 55 countries, representing 20 genera and 75 species collected from 1948 to 2023. Collections were performed in all regions of Brazil, comprising specimens from 26 states and the Federal District, particularly from the Southeast and Northeast. Within the dataset, *Biomphalaria* includes 3,926 lots of 31 species from 42 countries. These records will help improve our comprehension of schistosomiasis transmission dynamics and the geographic distributions of these medically important species.

**Submitted:** 15 July 2023

\* Corresponding author. E-mail: maribiorural@gmail.com

Preprint submitted at https://doi.org/10.5281/zenodo.10213911

Included in the series: ***Vectors of human disease*** (https://doi.org/10.46471/GIGABYTE_SERIES_0002)

**Subjects** Ecology, Biodiversity, Taxonomy

## DATA DESCRIPTION

### Background and context

The family Planorbidae includes around 40 genera of freshwater gastropods, with around 250 species widely distributed [1]. In this family, the subfamily Planorbinae includes six genera reported in Brazil: *Acrorbis*, *Antillorbis*, *Biomphalaria*, *Drepanotrema*, *Helisoma*, and *Plesiophysa* [2] (Figure 1). *Biomphalaria* includes species that act as intermediate hosts of the trematode *Schistosoma mansoni*, which causes schistosomiasis mansoni [3]. In Brazil, *S. mansoni* utilizes three species of the genus *Biomphalaria* as its natural intermediate hosts: *Biomphalaria glabrata* (Say, 1818), *B. straminea* (Dunker, 1848), and *Biomphalaria tenagophila* (d'Orbigny, 1835) [4, 5].

In this paper, we contributed a dataset derived from the Planorbidae species deposited in the Oswaldo Cruz Institute's Mollusk Collection, mainly from Brazil, but also from

**Figure 1.** Diversity of shell forms in Planorbidae from Brazil. (A) *Biomphalaria straminea* (CMIOC 5612), (B) *Drepanotrema lucidum* (CMIOC 5573), (C) *Helisoma duryi* (CMIOC 2318), (D) *Antillorbis nordestensis* (CMIOC 4550), (E) *Acrorbis petricola* (CMIOC 2744), (F) *Plesiophysa dolichomastix* (CMIOC 2041), and (G) *Gundlachia ticaga* (CMIOC 14689).

numerous other countries. This material was mainly the result of decades of study in freshwater ecosystems by Dr. Wladimir Lobato Paraense, known for his studies on the biology and taxonomy of Brazilian planorbids, and by his team in the laboratory of Malacology of the Oswaldo Cruz Institute. Many other specialists also contributed and examined several specimens from this collection. In 1948, when Dr. Wladimir Lobato Paraense worked at the Public Health Special Service, a Brazilian institution responsible for the control of parasitic diseases in the Rio Doce Valley in Minas Gerais, including schistosomiasis [6, 7], he began his studies on the mollusks involved in the transmission of schistosomiasis in Brazil, creating the collection.

Currently, this collection includes mainly freshwater and land gastropods involved in the transmission of other parasitic diseases, such as fascioliasis and angiostrongyliasis, both cerebral and abdominal, but also includes gastropod species that cause economic losses in agriculture (mainly exotic species) and among native species from the Brazilian Biomes. The collection contributes to science, research, and education. It also serves as a repository of knowledge about Brazilian and global mollusk biodiversity [6].

The datasets presented in this study consist of metadata associated with each batch of Planorbidae specimens, featuring varying numbers of specimens. We filled in the obligatory fields and have successfully passed screening using the integrated publishing toolkit (IPT) of

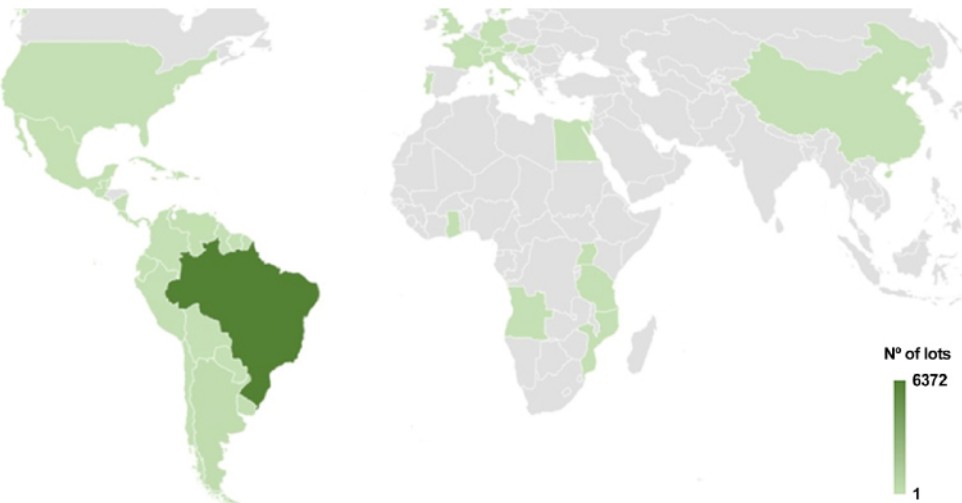

**Figure 2.** Spatial coverage of the occurrence dataset of Planorbidae, from 1948 to 2023, based on the CMIOC, including mainly lots are from Brazil (6,372 lots).

the Fundação Oswaldo Cruz (FIOCRUZ). For each lot of Planorbidae, our dataset includes fields providing, in Darwin Core Standard format, the following information: (i) taxonomy (kingdom, phylum, class, order, family, genus, specificEpithet, verbatimIdentification, infraspecificEpithet, scientificName, scientificNameAuthorship, taxonRank); (ii) collection details, including the collectors, collection date, collection site description (verbatimEventDate, eventTime, habitat, samplingProtocol); (iii) geolocation data (stateProvince, county, locality, locationRemarks, verbatimLatitude, verbatimLongitude, decimalLatitude, decimalLongitude, geodeticDatum); and (iv) catalog reference data (otherCatalogNumbers). This dataset is also available in the Sistema de Informação sobre a Biodiversidade Brasileira (SiBBr; i.e., Information System on the Brazilian Biodiversity), which integrates data and information, constituting the Brazilian Node of the Global Biodiversity Information Facility (GBIF), in an online platform for public use [8].

## METHODS

This study included all reports of Planorbidae genera and species from the dataset obtained from the Oswaldo Cruz Institute's Mollusk Collection (CMIOC). These lots are mainly from Brazil, but also include material from more than 50 countries and continents (Figure 2). The temporal coverage of the Planorbidae dataset is from 1948 to 2023.

All mollusks were morphologically identified in the laboratory to their genus and species based on shell and anatomical characteristics through specimen dissections, literature, and comparison with the lots deposited in the CMIOC [2, 9–12].

## DATA VALIDATION AND QUALITY CONTROL

Over the years, Dr. Lobato Paraense and his group published several works on planorbids from Brazil [9, 13–33]. Their works described and redescribed species of Planorbinae, reinforcing the importance of planorbids in transmitting diseases. Data validation was also done via the GBIF data-validator tool upon data submission [8].

**Table 1.** Genera and species of the Planorbidae family deposited in the CMIOC.

| Genus | Species |
|---|---|
| *Acrobis* | *Acrorbis petricola* |
| *Antillorbis* | *Antillorbis nordestensis, Antillorbis salleanus* |
| *Biomphalaria* | *Biomphalaria adowensis, Biomphalaria alexandrina, Biomphalaria amazonica, Biomphalaria andecola, Biomphalaria choanomphala, Biomphalaria costata, Biomphalaria cousini, Biomphalaria glabrata, Biomphalaria havanensis, Biomphalaria helophila, Biomphalaria intermedia, Biomphalaria kuhniana, Biomphalaria nicaraguana, Biomphalaria obstructa, Biomphalaria occidentalis, Biomphalaria oligoza, Biomphalaria orbignyi, Biomphalaria pallida, Biomphalaria peregrina, Biomphalaria pfeifferi, Biomphalaria prona, Biomphalaria pucaraensis, Biomphalaria schrammi, Biomphalaria sericea, Biomphalaria stanleyi, Biomphalaria straminea, Biomphalaria subprona, Biomphalaria sudanica, Biomphalaria tenagophila, Biomphalaria thermala, Biomphalaria trigyra.* |
| *Drepanotrema* | *Drepanotema anatinum, Drepanotrema beltrani, Drepanotrema cimex, Drepanotrema depressissimum, Drepanotrema heloicum, Drepanotrema kermatoides, Drepanotrema limayanum, Drepanotrema lucidum, Drepanotrema pfeifferi, Drepanotrema pileatum, Drepanotrema simonsi, Drepanotrema surinamense* |
| *Ferrissia* | *Ferrissia fragilis* |
| *Gundlachia* | *Gundlachia radiata, Gundlachia ticaga* |
| *Gyraulus* | *Gyraulus acronicus, Gyraulus albus, Gyraulus boetzkesi, Gyraulus crista, Gyraulus percarinatus* |
| *Hebetancylus* | *Hebetancylus moricandi* |
| *Helisoma* | *Helisoma anceps, Helisoma campanulatum, Helisoma caribaeum, Helisoma corpulentum, Helisoma duryi, Helisoma foveale, Helisoma peruvianum, Helisoma trivolvis* |
| *Hippeutis* | *Hippeutis complanatus* |
| *Laevapex* | *Laevapex diaphanous* |
| *Planorbarius* | *Planorbarius corneus* |
| *Planorbis* | *Planorbis boissyi, Planorbis canonicus, Planorbis corneus, Planorbis metidjensis, Planorbis planorbis, Planorbis salleanus* |
| *Segmentina* | *Segmentina nitida* |
| *Tropicorbis* | *Tropicorbis riisei* |
| *Uncancylus* | *Uncancylus concentricus* |

## RESULTS

The CMIOC has records of representatives of the family Planorbidae (Table 1) from 55 out of the 193 countries recognized by the United Nations in 2023, spanning across four out of the six continents. These countries include Germany, Angola, Antigua, Antigua and Barbuda, Argentina, Austria, Barbados, Belize, Bolivia, Brazil, Chile, China, Costa Rica, Colombia, Cuba, Egypt, El Salvador, Ecuador, United States of America, France, Ghana, Guadeloupe, Guatemala, Guyana, French Guiana, Haiti, Hong Kong, Hungary, England, Israel, Italy, Jamaica, Martinique, Mexico, Mozambique, Nicaragua, Panama, Paraguay, Peru, Puerto Rico, Portugal, Dominican Republic, Saint Croix, Saint Thomas, Saint Vincent, Saint Lucia, Sweden, Suriname, Tahiti, Tanzania, Trinidad, Uganda, Uruguay, and Venezuela. These reports cover the following coordinates: 90°0′0″S and 90°0′0″N Latitude; 180°0′0″W and 180°0′0″E Longitude.

In Brazil, the geographical distribution covers all five territorial regions, spanning the 26 states and the Federal District. The present database presents records of the occurrence of representatives of the family Planorbidae in a total of 592 municipalities (Midwest: recorded in 32 municipalities; Northeast: 120 municipalities; North: 27 municipalities; Southeast: 339 municipalities; and South: 74 municipalities). *Biomphalaria* is the most well-represented genus in CMIOC, including lots also from Africa, Asia, and Latin and North Americas, with a total of 3,926 lots. These specimens are registered in the following countries: Angola, Antigua, Argentina, Barbados, Belize, Bolivia, Chile, China, Costa Rica, Cuba, Egypt, El Salvador, Ecuador, United States, Ghana, Guadeloupe, Guatemala, Guyana, French Guiana, Haiti, Hong Kong, Jamaica, Martinique, Mexico, Mozambique, Nicaragua, Panama, Paraguay, Peru, Puerto Rico, Dominican Republic, Santa Lucia, Suriname, Tanzania, Trinidad, Uganda, Uruguay, and Venezuela (Table 2). Another well-represented genus of Planorbinae is *Drepanotrema*, which includes twelve species from different countries, with 2,312 lots (Table 3).

**Table 2.** Origin of the lots of *Biomphalaria* deposited in the CMIOC.

| Country | Species | Country | Species | Country | Species | Country | Species |
|---|---|---|---|---|---|---|---|
| Angola | *B. adowensis* | Ecuador | *B. cousini* | Hong Kong | *B. straminea* | Dominican Republic | *B. glabrata* |
| Antigua | *B. glabrata* | | *B. peregrina* | Jamaica | *B. helophila* | | *B. helophila* |
| Argentina | *B. intermedia* | | *B. sericea* | | *B. pallida* | | *B. straminea* |
| | *B. oligoza* | | *B. trigyra* | Martinique | *B. glabrata* | Saint Lucia | *B. glabrata* |
| | *B. orbignyi* | United States of America | *B. glabrata* | | *B. kuhniana* | Suriname | *B. glabrata* |
| | *B. peregrina* | | *B. havanensis* | | *B. straminea* | | *B. kuhniana* |
| | *B. straminea* | | *B. peregrina* | Mexico | *B. obstructa* | | *B. straminea* |
| | *B. tenagophila* | | *B. obstructa* | Mozambique | *B. pfeifferi* | Tanzania | *B. choanomphala* |
| Barbados | *B. helophila* | Ghana | *B. pfeifferi* | Nicaragua | *B. helophila* | | *B. pfeifferi* |
| Belize | *B. helophila* | Guadeloupe | *B. glabrata* | | *B. nicaraguana* | | *B. sudanica* |
| | *B. obstructa* | | *B. kuhniana* | | *B. obstructa* | Trinidad | *B. straminea* |
| Bolivia | *B. andecola* | | *B. schrammi* | Panama | *B. kuhniana* | Uganda | *B. stanleyi* |
| | *B. pucaraensis* | Guatemala | *B. helophila* | Paraguay | *B. peregrina* | Uruguay | *B. straminea* |
| Chile | *B. costata* | | *B. obstructa* | | *B. occidentalis* | | *B. tenagophila* |
| | *B. peregrina* | | *B. subprona* | | *B. tenagophila* | | *B. tenagophila guaibensis* |
| | *B. thermala* | Guyana | *B. glabrata* | | *B. straminea* | Venezuela | *B. glabrata* |
| China | *B. straminea* | | *B. schrammi* | Peru | *B. andecola* | | *B. peregrina* |
| Costa Rica | *B. helophila* | | *B. straminea* | | *B. helophila* | | *B. prona* |
| | *B. straminea* | French Guiana | *B. glabrata* | | *B. peregrina* | | *B. straminea* |
| Cuba | *B. havanensis* | Haiti | *B. glabrata* | | *B. pucaraensis* | | |
| | *B. helophila* | | *B. havanensis* | | *B. tenagophila* | | |
| Egypt | *B. alexandrina* | | *B. helophila* | | *B. trigyra* | | |
| El Salvador | *B. helophila* | | *B. pallida* | Puerto Rico | *B. glabrata* | | |
| | *B. obstructa* | | *B. obstructa* | | *B. helophila* | | |
| | | | *B. straminea* | | *B. peregrina* | | |

Considering lots from Brazil, the collection at CMIOC includes representatives from all 11 species of *Biomphalaria* that occur nationwide. Among them, three species play a significant role in the biological cycle of Brazil [33] and have a wide distribution. Together, these species account almost 40% of the total Planorbidae lots in the collection. Specifically, *B. straminea* is the most represented species with 1,257 lots, while *B. tenagophila*, and *B. glabrata* account for 811 and 654 lots, respectively.

The three species of *Biomphalaria* found in Brazil and recorded at CMIOC exhibit distinct geographic distributions within the country. However, there are certain states where these species overlap, indicating areas of coexistence (Figures 3–5).

*B. straminea* has the broadest geographic range among the three species in Brazil. It can be found in various states, from the North to the South of the country. Its distribution encompasses regions such as the Amazon, the Cerrado, the Caatinga, and the Atlantic Forest. The extensive presence of *B. straminea* highlights its significance as an intermediate host for the parasitic trematode. For a complete record of *B. straminea* in Brazil [34], CMIOC only lacks samples from the states of Santa Catarina and Roraima (Figure 3). On the other hand, *B. tenagophila* has a more restricted distribution compared to *B. straminea. B. tenagophila* is commonly found in areas of the South and Southeast of Brazil, primarily encompassing the states of São Paulo, Paraná, Santa Catarina, and Rio Grande do Sul (Figure 4). Finally, *B. glabrata* has a less limited geographic distribution than *B. tenagophila* in Brazil. Specifically, *B. glabrata* is predominantly found in coastal areas, particularly in the Northeast region of the country (Figure 5).



**Table 3.** Origin of the lots of *Drepanotrema* deposited in the CMIOC.

| Species | Country | Number of lots | Species | Country | Number of lots |
|---|---|---|---|---|---|
| *Drepanotrema anatinum* | Argentina | 11 | *Drepanotrema kermatoides* | Argentina | 22 |
| | Belize | 1 | | Brazil | 32 |
| | Brazil | 521 | | Ecuador | 2 |
| | Costa Rica | 1 | | Paraguay | 1 |
| | Ecuador | 1 | | Peru | 10 |
| | Guatemala | 3 | | Uruguay | 12 |
| | Guyana | 1 | *Drepanotrema limayanum* | Peru | 6 |
| | Haiti | 2 | *Drepanotrema pfeifferi* | Argentina | 1 |
| | Jamaica | 4 | | Chile | 3 |
| | Mexico | 1 | *Drepanotrema pileatum* | Brazil | 8 |
| | Nicaragua | 1 | *Drepanotrema lucidum* | Antigua | 1 |
| | Panama | 1 | | Argentina | 23 |
| | Puerto Rico | 5 | | Barbados | 1 |
| | Dominican Republic | 2 | | Belize | 2 |
| | Saint Lucia | 1 | | Brazil | 808 |
| | Suriname | 2 | | Ecuador | 2 |
| | Trinidad | 3 | | United States | 1 |
| | Uruguay | 1 | | Guadeloupe | 4 |
| | Uruguay | 2 | | Guatemala | 1 |
| | Venezuela | 2 | | Haiti | 1 |
| *Drepanotrema beltrani* | Mexico | 1 | | Jamaica | 8 |
| *Drepanotrema cimex* | Argentina | 7 | | Mexico | 3 |
| | Brazil | 369 | | Nicaragua | 2 |
| | Haiti | 1 | | Paraguay | 5 |
| | Jamaica | 3 | | Puerto Rico | 6 |
| | Puerto Rico | 1 | | Dominican Republic | 2 |
| | Uruguay | 3 | | Saint Vincent | 1 |
| | Venezuela | 3 | | Saint Lucia | 3 |
| *Drepanotrema depressissimum* | Antigua | 1 | | Uruguay | 4 |
| | Argentina | 13 | | Trinidad | 2 |
| | Barbados | 2 | | Venezuela | 1 |
| | Brazil | 295 | *Drepanotrema simonsi* | Puerto Rico | 1 |
| | Costa Rica | 1 | *Drepanotrema surinamense* | Costa Rica | 2 |
| | Guadeloupe | 7 | | Ecuador | 2 |
| | Nicaragua | 2 | | Guadeloupe | 1 |
| | Paraguay | 1 | | Guyana | 1 |
| | Peru | 1 | | Panama | 4 |
| | Saint Lucia | 2 | | Suriname | 4 |
| | Uruguay | 3 | | | |
| | Venezuela | 1 | | | |
| *Drepanotrema heloicum* | Argentina | 15 | | | |
| | Brazil | 3 | | | |
| | Uruguay | 8 | | | |

## DATA VALIDATION AND QUALITY CONTROL

Planorbidae specimens were identified by experienced taxonomists. The dataset is in Darwin Core format, and all mandatory fields are present and have undergone screening in the FIOCRUZ IPT.

## REUSE POTENTIAL

The presented dataset is important because it provides information on the distribution of Planorbidae, Planorbinae, and *Biomphalaria* in Brazil based on a renowned collection of medical malacology (i.e., CMIOC), traditionally known for its studies within the country.



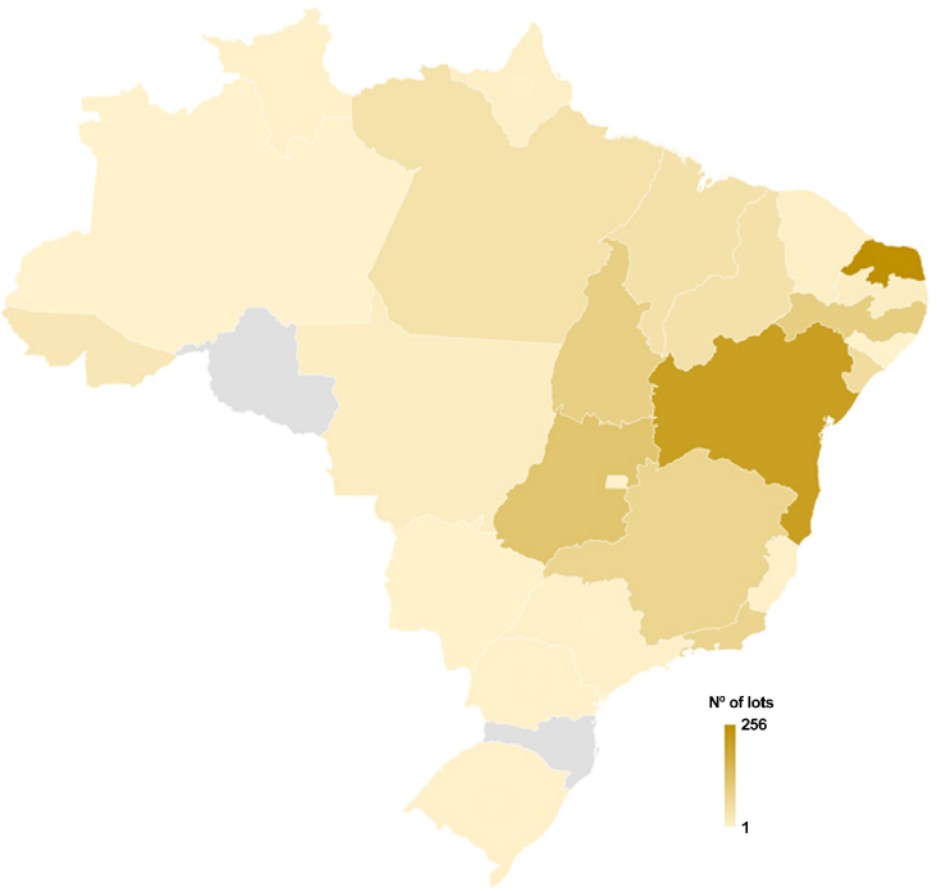

**Figure 3.** Spatial coverage of the occurrence dataset of *B. straminea*, from 1948 to 2023, based on the CMIOC, showing that most lots are from Rio Grande do Norte, with 256 lots.

This dataset can provide the basis for future studies in evolution, ecology, and epidemiology, among others, especially for species of medical interest from the public health perspective. An important point of the current collection is the first recording of *B. straminea* in Amapá, serving as an important reference for research into the biodiversity of Planorbinae. These data expand the distribution of these species and provide occurrence information on the other species of this genus in Brazil. In addition to supporting the surveillance and control of schistosomiasis in Brazil, these data also contribute to the knowledge of *Biomphalaria* biodiversity. They are also an important resource for managing the CMIOC (Figure 6).

## DATA AVAILABILITY

The dataset used in this article is published through the FIOCRUZ – Oswaldo Cruz Foundation IPT – and is provided under a CC0 waiver from GBIF [8] and in the SiBBr repository [34].

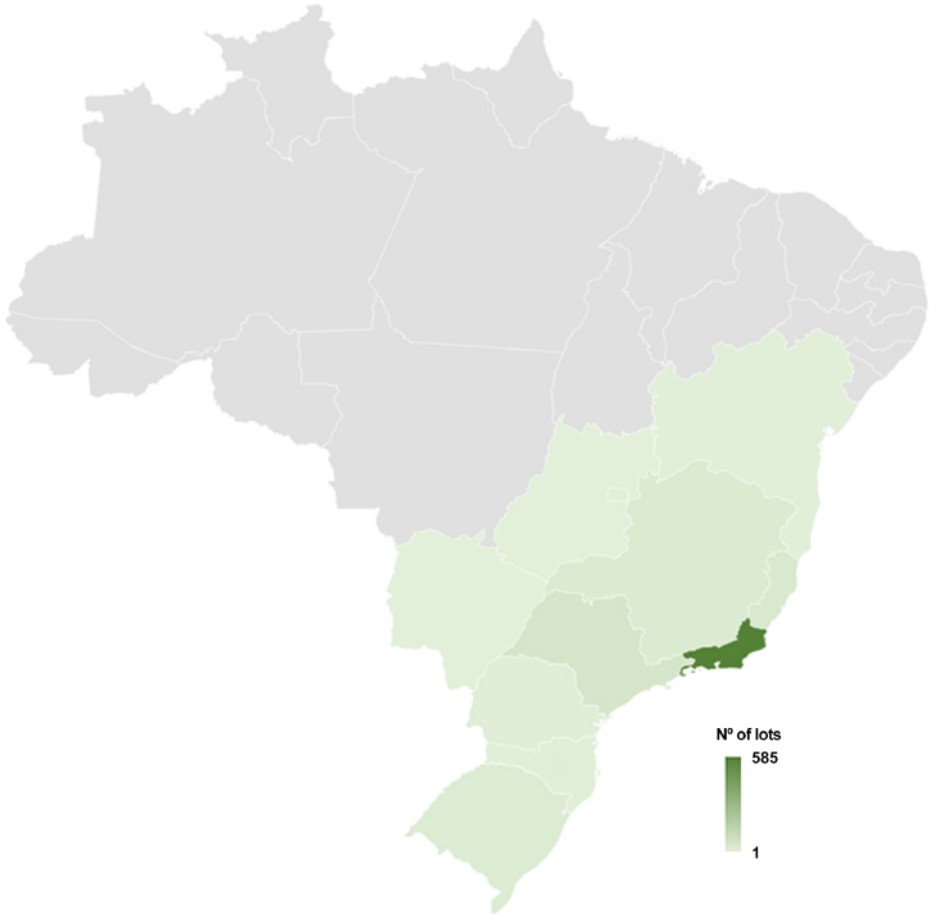

**Figure 4.** Spatial coverage of the occurrence dataset of *B. tenagophila*, from 1948 to 2023, based on the CMIOC, showing that most lots are from Rio de Janeiro State, with a total of 585 lots. The second more represented State is São Paulo, with 54 lots.

## EDITOR'S NOTE

This paper is part of a series of Data Release articles working with GBIF and supported by TDR, the Special Programme for Research and Training in Tropical Diseases hosted at the World Health Organization, in order to publish datasets on vectors of human diseases [35].

## ABBREVIATIONS

CMIOC, Collection of Mollusks of the Oswaldo Cruz Institute; FIOCRUZ, Fundação Oswaldo Cruz; GBIF, Global Biodiversity Information Facility; IPT: integrated publishing toolkit; SiBBr, Sistema de Informação sobre a Biodiversidade Brasileira.

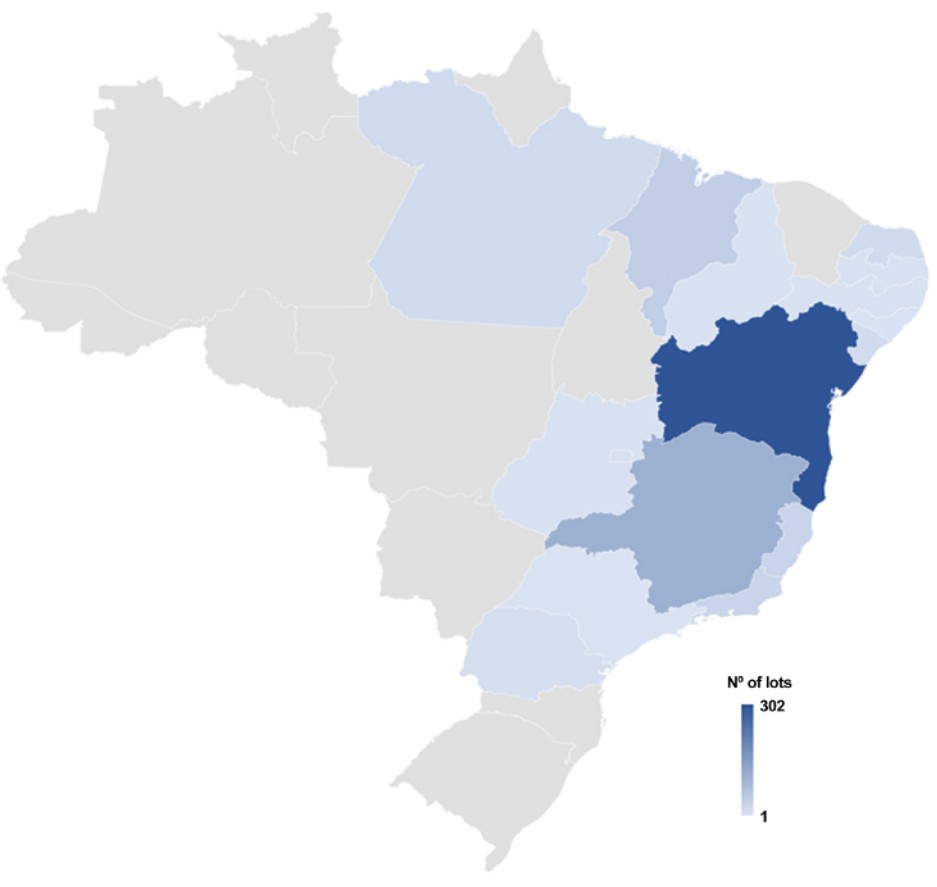

**Figure 5.** Spatial coverage of the occurrence dataset of *B. glabrata*, from 1948 to 2023, based on the CMIOC, showing that most lots are from Bahia, with 302 lots.

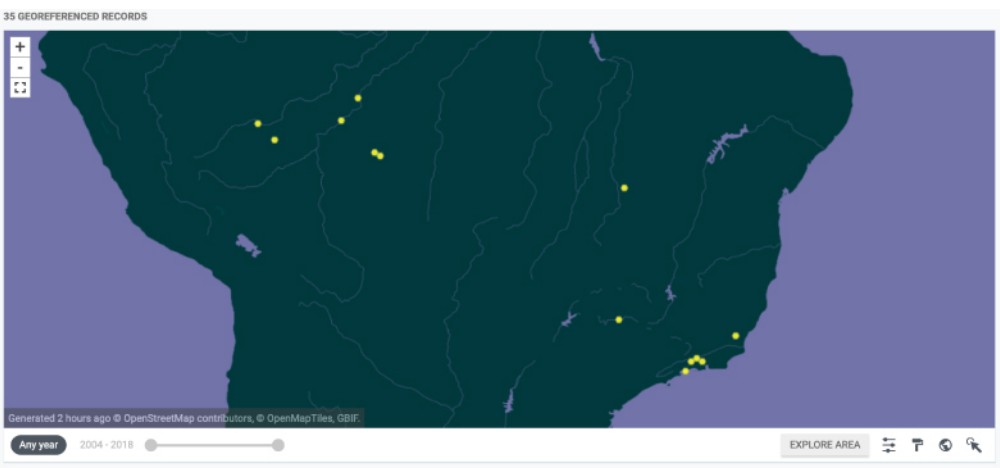

**Figure 6.** Interactive map of the georeferenced occurrences hosted by GBIF [8].
https://www.gbif.org/dataset/2bd86564-44c0-4317-a04e-79b544a84a06

## DECLARATIONS

### Ethical approval

The authors declare that ethical approval was not required for this type of research.

### Competing interests

The authors declare that they have no competing interests.

### Funding

Collections of the lots of the studied period were funded by numerous institutions, including World Health Organization (WHO), Pan American Health Organization (PAHO), Brazilian Health Ministry (MS), Oswaldo Cruz Foundation, FURNAS Centrais Elétricas by Centrais Elétricas Brasileiras S.A. - Eletrobras, Fundação de Amparo à Pesquisa do Estado do Rio de Janeiro (FAPERJ), Conselho Nacional de Desenvolvimento Científico e Tecnológico (CNPq).

### Acknowledgements

We thank Clara Baringo Fonseca (SiBBr - Rede Nacional de Ensino e Pesquisa - Rnp) for their support during the dataset preparation and Paloma Helena Fernandes Shimabukuro from GBIF for their encouragement while we developed this study. We also thank all CMIOC staff for their contribution over the years.

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
