## [Editor Report]

Editor’s AssessmentPlanorbidae are a family of air-breathing freshwater snails that include species that act as natural intermediate hosts for the trematode Schistosoma mansoni Sambon, which causes schistosomiasis in humans and animals. This work presents a dataset of Planorbidae snails housed in the Mollusc Collection of the Oswaldo Cruz Institute (CMIOC) in Brazil, particularly targeting the Biomphalaria species that carry disease. This consists of 7,268 lots, originating from 53 different countries, representing 16 genera and 61 species, and all collected from 1948 to 2023. After some work reviewing and curating this dataset, it has been published and shared as a Darwin Core Archive via the GBIF database. These records hopefully aiding the understanding of schistosomiasis transmission dynamics and geographic distributions of these medically important species.

---

## [Reviewer Report]

Upload additional filesDRR-202307-02/form/DRR-202307-02_Data-Review-MAT (1).pdfReviewer name and names of any other individual's who aided in reviewer Mary Ann TuliDo you understand and agree to our policy of having open and named reviews, and having your review included with the published papers. (If no, please inform the editor that you cannot review this manuscript.)YesIs the language of sufficient quality?YesPlease add additional comments on language quality to clarify if needed
GBIF record is in Portuguese. Are all data available and do they match the descriptions in the paper? NoAdditional CommentsFrom the data review document: The Description (it’s in Portuguese) in GBIF talks about the whole mollusc collection rather than just the 7268 item dataset, which is the subject of the manuscript and what the Occurence dataset is comprised of. It also points to http://cmioc.fiocruz.br/ which is a top level website. I could not find how to navigate to the 7268 item dataset from this page. Clarification is needed. 
Are the data and metadata consistent with relevant minimum information or reporting standards? See GigaDB checklists for examples <a href="http://gigadb.org/site/guide" target="_blank">http://gigadb.org/site/guide</a>YesAdditional CommentsIs the data acquisition clear, complete and methodologically sound?YesAdditional CommentsIs there sufficient detail in the methods and data-processing steps to allow reproduction?YesAdditional CommentsIs there sufficient data validation and statistical analyses of data quality? YesAdditional CommentsIs the validation suitable for this type of data?YesAdditional CommentsIs there sufficient information for others to reuse this dataset or integrate it with other data?YesAdditional CommentsAccessibility of the Occurrence dataset via GBIF is comprehensive, clear and easy to navigate and use.Any Additional Overall Comments to the AuthorThe following section starting "The CMIOC holds deposits of 61 species and 16 genera from the family Planorbidae:Acrobis(Acrorbis petricola), ....." could be moved to a table; it is awkward to read as part of the body of the text.  Minor points from the dataset review: 1 - GBIF publication data is BEFORE registration date.  2 - Typo in manuscript. Figure 3. Biomphalaria glabrata occurrence dataset - > Figure5. Biomphalaria glabrata occurrence dataset. nb appears just before DATA VALIDATION section. 
RecommendationMinor Revision

---

## [Reviewer Report]

Reviewer name and names of any other individual's who aided in reviewer Tom PennanceDo you understand and agree to our policy of having open and named reviews, and having your review included with the published papers. (If no, please inform the editor that you cannot review this manuscript.)YesIs the language of sufficient quality?YesPlease add additional comments on language quality to clarify if needed
Correct to 'Species' in Table 1. Correct to 'contributed' in background and introduction.Are all data available and do they match the descriptions in the paper? YesAdditional CommentsAre the data and metadata consistent with relevant minimum information or reporting standards? See GigaDB checklists for examples <a href="http://gigadb.org/site/guide" target="_blank">http://gigadb.org/site/guide</a>YesAdditional CommentsIs the data acquisition clear, complete and methodologically sound?YesAdditional CommentsIs there sufficient detail in the methods and data-processing steps to allow reproduction?YesAdditional CommentsIs there sufficient data validation and statistical analyses of data quality? Not my area of expertiseAdditional CommentsIs the validation suitable for this type of data?YesAdditional CommentsIs there sufficient information for others to reuse this dataset or integrate it with other data?YesAdditional CommentsAny Additional Overall Comments to the AuthorI was asked to review the collection processes and methods specifically, for which I see no issues with. My only comment would be that it seems an oversight in the background of the dataset to not mention Bulinus as also being in the planorbidae family and transmitting schistosomiasis, but that these are not present in Brazil (just for sake of clarity). Also - correct to 'Species' in Table 1. Correct to 'contributed' in background and introduction. In Table 1, I would also like to see it reformatted so that there is a gap or bold border between 'Country' and 'Species' so that it is clear that columns are repeated across the table.RecommendationAccept